# OpenReview forum: "SelfJudge: Faster Speculative Decoding via Self-Supervised Judge Verification"
_ICML.cc/2026/Conference — ICML 2026 regular_

### Official Review · Reviewer_eQ6q · 2026-03-03

**Soundness:** 3
**Presentation:** 2
**Significance:** 3
**Originality:** 3
**Overall Recommendation:** 5
**Confidence:** 4

**Summary:**

This work presents SelfJudge, a judge verification framework for Speculative Decoding (SD) that accelerates LLM inference without human annotations or task-specific ground truths. Standard SD uses strict alignment-based verification that conservatively rejects draft tokens even when they preserve semantic meaning. Prior "judge" methods (e.g., AutoJudge) relaxed this but required human labeling or were limited to math/code tasks. SelfJudge overcomes this via self-supervision: it computes a "semantic preservation score" measuring the log-likelihood difference between the target model's original response and a token-substituted variant using both prefix and suffix contexts. A lightweight logistic regression verifier is trained on these self-generated labels. Evaluations on GSM8K, MATH-500, MMLU, CNN/DM, and LiveCodeBench show SelfJudge achieves +2.06 average accepted tokens per cycle with only -1.0% accuracy drop compared to standard SD.

**Compliance With Llm Reviewing Policy:**

Affirmed.

**Final Justification:**

My concerns have been adequately addressed. I have increased my overall score from 4 to 5.

**Key Questions For Authors:**

## Questions for Authors

1. **Regarding Verifier Complexity:** Appendix D.3 shows the logistic regression verifier adds only 0.02s on a V100 GPU, which is impressively low. Given this headroom, did you experiment with slightly more expressive, non-linear verifiers (e.g., small MLPs or attention blocks) that might better capture the bidirectional semantics of your training labels? Would such a verifier improve acceptance accuracy while still maintaining negligible overhead?
2. **Regarding Threshold Calibration and Performance Ceiling:** Your method beautifully avoids human annotation via self-supervised semantic preservation scoring. Yet, the threshold $\tau$ is ultimately calibrated to strictly adhere to AutoJudge's rejection labels on 100 GSM8K samples. If SelfJudge is explicitly tuned to emulate AutoJudge’s conservatism on this calibration set, doesn't that inherently cap SelfJudge’s capacity or limit its performance to be bounded by AutoJudge's capabilities? How would you design a purely autonomous calibration method that removes this final dependency on external/prior verifiers?
3. **Regarding Out-of-Domain Generalization:** Your training data (GSM8K, LiveCodeBench, Dolly) covers math, code, and general instructions, which aligns closely with the evaluated benchmarks. How well does a verifier trained on this mix generalize to a *completely unseen* domain (e.g., clinical notes, dense legal contracts)? Does a new target domain require a completely new self-supervised training phase to maintain safety and speedup?

**Limitations:**

yes

**Strengths And Weaknesses:**

## Strengths

* **Scalable Self-Supervised Data Generation:** The most impactful contribution of this paper is the self-supervised data generation pipeline. Prior judge decoding methods relied on either expensive human annotations (JudgeDecoding) or task-specific ground truths limited to math/code (AutoJudge). SelfJudge completely eliminates both requirements by using the target model itself to label tokens, providing a clean, automated, and scalable pipeline that can be applied to any target model without domain constraints.
* **Principled Bidirectional Semantic Preservation Score:** The introduction of the bidirectional semantic preservation score is mathematically sound. As detailed in the Bayesian interpretation in Section 3.3.1, incorporating the suffix likelihood (Log Bayes Factor) provides much stronger evidence for semantic equivalence than simply looking at prefix probabilities. This principled formulation elevates the verification criterion from simple probability comparison to an evidence-based update.
* **Strong Empirical Results in the Family of Relaxed Speculative Decoding:** The comprehensive evaluation (Table 1) clearly demonstrates SelfJudge's superiority over Top-k, AutoJudge-R, and AutoJudge-F on the tested benchmarks. SelfJudge achieves speedups on MMLU and CNN/DM—where AutoJudge showed almost no improvement or actively degraded performance. Overall, this is a good contribution to the emerging family of relaxed speculative decoding methods that trade strict distributional equivalence for practical speedup, and demonstrates a promising direction for making such methods more broadly applicable.

## Weaknesses

* **Verifier Architecture Mismatch with Training Signal:** The paper's core insight is that bidirectional context (prefix + suffix) is essential for accurate semantic preservation labeling. However, during inference, the verifier is a simple logistic regression operating only on the current token's hidden state ($h_t$). Appendix D.3 shows this lightweight choice adds only 0.02s (0.26% of total verification time) on a V100 GPU, making it practically negligible. However, this near-zero overhead comes at the cost of expressiveness — a linear classifier on a single hidden state cannot fully capture the bidirectional semantic complexity used to generate its own training labels. The paper does not explore whether slightly more expressive verifiers (e.g., small MLPs) could improve acceptance accuracy while still maintaining acceptable overhead.
* **Threshold Calibration Bottlenecked by AutoJudge:** A key claim of SelfJudge is its independence from human annotations and task-specific datasets, relying fully on a self-supervised pipeline. However, Section 4.1 details that the critical threshold $\tau$ (which determines semantic preservation) is calibrated using 100 sampled queries from GSM8K against AutoJudge's ground-truth token labels. By forcing SelfJudge’s threshold to match AutoJudge’s rejection decisions on these samples, it implies SelfJudge's fundamental capability might be conceptually "capped" by AutoJudge's performance ceiling, at least on the calibration domain. This external dependency partially undermines the narrative of a completely autonomous, self-supervised verifier.
* **Unclear Out-of-Domain Generalization:** The authors train the verifier using self-supervised labels generated from GSM8K (math), LiveCodeBench (code), and Dolly15k (instruction tuning). They then evaluate on GSM8K, MATH-500, MMLU, CNN/DM, and LiveCodeBench. Given the significant overlap in task types between the training prompts and the evaluation benchmarks, it is difficult to determine how well the verifier generalizes to truly out-of-domain distributions (e.g., highly specialized legal, medical, or creative text) without generating new task-specific training data.

---

> ### Author Rebuttal · Authors · 2026-03-30
>
> ### W1 & Q1: Investigate Verifier with Complex Architecture.
> ---
>
> We agree that verifier architecture is an important design dimension. In this work, we intentionally adopted logistic regression following JudgeDecoding and AutoJudge to establish **a fair comparison**, isolating the effect of our semantic preservation criterion from architectural choices.
> Hence, we conduct additional experiments with 2-layer and 3-layer MLP verifiers (ReLU activations).
>
> | Method         | Verifier   | GSM8K (m/Acc) | MMLU (m/Acc) |
> |---|---|:---:|:---:|
> | SD                | NA          |  9.14 / 80.7 | 4.36 / 65.0  |
> | SelfJudge-R | Logistic   |10.09 / 80.7 | 5.14 / 64.4  |
> | SelfJudge-R | MLP-2    | 10.33 / 80.6 | 5.40 / 64.7  |
> | SelfJudge-R | MLP-3    | 10.12 / 81.8 | 5.23 / 64.6  |
>
> In Table, we observed that **more expressive verifier architectures slightly improve SelfJudge's performance.** Importantly, as the logistic verifier accounts for only 0.26% of total verification time, and MLP verifiers also remain under 1%, adding negligible overhead. Hence, using more complex verifier than logistic regression would be better for SelfJudge. We thank the reviewer for this suggestion — exploring more expressive verifier architectures is a promising direction for further improving SelfJudge, and we will include a more thorough investigation in the revised version.
>
> ---
> ### W2 & Q2: Regarding $\tau$ Selection
> ---
>
> Please note that our $\tau$ setting based on AutoJudge is **not chosen for optimal performance, but for practicality**—it reduces the cost of parameter search. To set $\tau$ in a fully self-contained manner, we propose and validate a simple quantile-based calibration that requires no external method or ground truth: given the distribution of semantic preservation scores computed by SelfJudge itself across training data (GSM8K train + LiveCodeBench train + Dolly15k), **we can select $\tau$ as a quantile of this distribution.** This is purely endogenous—it depends only on the target model's own likelihood assessments.
> To further support the robustness of SelfJudge, we provide a sensitivity analysis on $\tau$ by varying it according to quantiles $\{10, 30, 50, 70, 80, 90\}$ of this distribution. Q75* denotes the $\tau$ used in our paper, which was calibrated via AutoJudge.
>
>
> |Quantile | Q10 | Q30 | Q50|Q70|Q75* |Q80 |Q90|
> |---|---|---|---|---|---|---|---|
> | $\tau$     | 0.335 | 0.193 | 0.126 | 0.076 | 0.065 | 0.052 | 0.025|
> | m 	      | 13.71 | **12.10** | 11.41 | 10.26  | 10.09  | 9.87   | 9.72
> | Acc.	      | 77.9   | **80.7**   | 80.5   |  80.4    | 80.7  | 80.7  |  81.0
>
> In the table above, we report results on GSM8K (test) and make two key observations:
> - First, **performance remains stable** across a wide range of $\tau$, which allows practitioners to select the hyperparameter without extensive tuning.
> - Second, our chosen setting (based on answer-preservation via AutoJudge) **does not even yield the best performance** of our method. SelfJudge has a room for improving its performance by tuning hyperparameter $\tau$. This confirms that our hyperparameter choice in the manuscript is strictly conservative, yet still outperforms baselines, demonstrating its practical robustness.
>
> Given these results, we propose the following fully endogenous calibration procedure: compute semantic preservation scores on diverse prompts using SelfJudge, then set τ\tau τ at a moderate quantile (e.g., Q30–Q90) of the score distribution. This removes all dependency on AutoJudge or any task-specific ground truth while preserving stable performance.
>
>
> ---
> ### W3 & Q3: Generalization on Out-of-Domain Task
> ---
>
> To further validate generalizability, we conduct experiments on LegalBench [1] and MedQA [2] — domains entirely absent from verifier training. In this experiment, we use the same verifier in our manuscript, which is trained on GSM8K, LiveCodeBench and Dolly15k.
>
> | Method    | LegalBench (m/Acc) | MedQA (m/Acc) |
> |---   |:---:|:---:|
> | SD                 | 4.14 / 4.7|  4.65 / 58.6 |
> | AutoJudge-F | 4.53 / 3.9 | 6.93 / 46.1 |
> | SelfJudge-R  | 4.56 / 4.8 | 6.18 / 57.7 |
> | SelfJudge-F  | 5.16 / 4.5 | 7.38 / 54.8  |
>
> In the above table, we observed that our proposed method generalizes well compared to SD and AutoJudge, indicating that our approach **generalizes well even if the verifier has never seen the domain** like Legal and Medical questions. This confirms that our semantic preservation criterion transfers to unseen domains far more effectively than answer-preservation-based approaches.
>
> [1] https://huggingface.co/datasets/nguha/legalbench
> [2] https://huggingface.co/datasets/bigbio/med_qa

---

> > ### Author Rebuttal · Reviewer_eQ6q · 2026-04-01
> >
> > My concerns have been adequately addressed. I have increased my overall score from 4 to 5.

---

> > > ### Author Response · Authors · 2026-04-01
> > >
> > > Thank you for confirming that your concerns have been fully resolved! We appreciate you taking the time to review our work.

---

### Official Review · Reviewer_F2Ab · 2026-03-08

**Soundness:** 3
**Presentation:** 2
**Significance:** 2
**Originality:** 3
**Overall Recommendation:** 5
**Confidence:** 2

**Summary:**

The paper proposes SelfJudge, a speculative decoding framework that trains a judge verifier using self-supervision from the target model (there’s no need for human annotations). The main idea is to measure semantic preservation by comparing token-substituted responses against the original under the target model's likelihood.

**Compliance With Llm Reviewing Policy:**

Affirmed.

**Final Justification:**

I've increased my final score to 5. The authors addressed my initial concerns in the rebuttal.

**Key Questions For Authors:**

1. Can you provide statistical significance tests or confidence intervals for the main results? Some differences between methods are small enough that it's hard to tell if they're meaningful.

2. The threshold \tau is calibrated using AutoJudge's labels on GSM8K problems. Did you explore other ways of setting this, and how sensitive are the results to this choice?

3. You use basic logistic regression for recognizing semantic-preserved tokens. This is a very weak classifier and the paper doesn't justify this choice or explore alternatives. Do you think you're leaving performance on the table? Have you experimented with other architectures?

4. How sensitive are the results to the choice of draft model?

**Limitations:**

Yes.

**Strengths And Weaknesses:**

Soundness: The method is well-motivated and the experiments are fairly comprehensive, but I still have some concerns. In particular, the threshold \tau is calibrated using AutoJudge’s labels on GSM8K problems, which introduces a dependency on task-specific ground truth that somewhat contradicts the “self-supervised” framing.

Presentation: I think the paper would benefit from a reorganization. The main motivation is inference speed, but you only report wall-clock speedups in Appendix D. Is this Appendix even cited in the main paper? Since the verifier adds overhead, this is critical information, and the results look good so there's no reason to hide them.

Significance: I think the paper can have practical impact. Faster inference is a real need and this approach makes sense. However, the paper lacks statistical significance tests, which matters given that some differences between methods are quite small. It is hard to be fully confident in the results without this.

Originality: The judge verification framework itself builds heavily on prior work, but the idea seems novel enough to me.

---

> ### Author Rebuttal · Authors · 2026-03-30
>
> ### Q1: Statistical Significance Test
> ---
>
> Thank you for raising this point. We conducted paired t-tests (5 runs with different random seeds) to verify the statistical significance of our results. Since the main results (Table 1) were obtained with temperature 0, which is deterministic and does not allow for variance estimation, we re-ran the experiments with temperature parameters 0.3 to introduce sampling variance across runs. We compare SelfJudge-F with AutoJudge-R, which achieves the best task performance while maintaining reasonable efficiency.
>
>
> **Accept length (m).**  In the below Table, we report mean ± std. It shows that SelfJudge-F significantly outperforms in terms of the averaged accept length when considering the standard deviation together. Moreover, the difference in accept length between the two methods is statistically significant across all four benchmarks (p < 0.001): GSM8K (t(4) = −21.46, p < 0.001), MATH-500 (t(4) = −13.30, p < 0.001), MMLU (t(4) = −37.55, p < 0.001), and CNN/DM (t(4) = −31.12, p < 0.001).
>
> | Method | GSM8K | MATH-500 | MMLU | CNN/DM | Avg. |
> |---|:---:|:---:|:---:|:---:|:---:|
> | AutoJudge-R |  9.64 ± 0.12 | 11.09 ± 0.11 | 4.33 ± 0.14 | 4.26 ± 0.09 | 7.33 |
> | SelfJudge-F   | 11.06 ± 0.11 | 12.37 ± 0.30 | 6.28 ± 0.12 | 5.82 ± 0.12 | 8.88 |
>
> **Task performance (Accuracy / FC).** The accuracy differences between AutoJudge-R and SelfJudge-F remain small across all benchmarks, confirming that the latency gains of SelfJudge-F do not come at a meaningful cost to task performance.
>
> | Method | GSM8K | MATH-500 | MMLU | CNN/DM | Avg. |
> |---|:---:|:---:|:---:|:---:|:---:|
> | AutoJudge-R | 79.6 ± 0.33 | 40.2 ± 0.48 | 64.1 ± 0.29 | 63.5 ± 0.66 | 61.8 |
>  | SelfJudge-F |  79.9 ± 0.35 | 41.6 ± 0.32 | 63.0 ± 0.41 | 63.8 ± 0.40 | 62.1 |
>
> These results imply that SelfJudge-F achieves significantly higher accept length than AutoJudge-R (p < 0.001 on all benchmarks) while maintaining comparable task performance, demonstrating a clear efficiency gain without sacrificing quality.
>
>
> ---
> ### Q2: Regarding $\tau$ Selection
> ---
>
> Due to the character limit of ICML rebuttal, we kindly ask the reviewer to read [Reviewer eQ6q](https://openreview.net/forum?id=9DAX4oLVhy&noteId=O5anWjgJXq) **W2&Q2.**
>
> ---
> ### Q3: Investigate Verifier with Complex Architecture.
> ---
>
> Due to the character limit of ICML rebuttal, we kindly ask the reviewer to read [Reviewer eQ6q](https://openreview.net/forum?id=9DAX4oLVhy&noteId=O5anWjgJXq) **W1&Q1.**
>
> ---
> ### Q4: Discussion on Different Draft Model
> ---
>
> In standard SD practice, draft models are selected from the same model family as the target model to maximize token alignment (Leviathan et al., 2023) [1]. This is not specific to SelfJudge but a well-established convention across the SD literature — Llama-3.1-8B is paired with Llama-3.2-1B, Qwen-2.5-7B with Qwen-2.5-0.5B, and so on. Our evaluation follows this standard protocol.
>
> Within this standard setting, SelfJudge's sensitivity to the draft model is structurally limited. The draft model only affects Step 1 of our pipeline — identifying mismatched tokens. The semantic preservation scores, training labels, and the verifier itself are all computed entirely from the target model's own likelihoods and hidden representations. This means the draft model determines which tokens are evaluated, but not how they are evaluated. A stronger draft model produces fewer mismatches (thus fewer training examples), while a weaker one produces more — but the labeling criterion remains the same.
>
> Our experiments span draft models with substantially different alignment characteristics: Llama-3.2-1B (1B for 8B target), Llama-3.1-8B (8B for 70B target), and Qwen-2.5-0.5B (0.5B for 7B target). Across all pairings, SelfJudge consistently outperforms baselines (Tables 1–2, Figure 3), suggesting SelfJudge can achieve speedup across various target-draft model pairs.
>
> ---
> ### Paper Revision: On wall-clock speedup placement on manuscript
> ---
>
> We fully agree with the reviewer. The wall-clock results (Tables 7–8) demonstrate substantial real speedups and there is no reason to relegate them to the appendix. We will move the wall-clock evaluation into the main paper in the revised version and ensure it is properly cited.
>
> [1] Fast Inference from Transformers via Speculative Decoding

---

> > ### Author Rebuttal · Reviewer_F2Ab · 2026-04-01
> >
> > Thanks for the rebuttal. I've increased my score to 5. Please make sure to move the wall-clock evaluation to the main paper in the revised version.

---

> > > ### Author Response · Authors · 2026-04-01
> > >
> > > We're glad we could address your concerns through the rebuttal process! We will incorporate the wall-clock evaluation into the main paper in the revised version as suggested. Thank you for your time and constructive feedback.

---

### Official Review · Reviewer_7AMQ · 2026-03-13

**Soundness:** 4
**Presentation:** 4
**Significance:** 2
**Originality:** 2
**Overall Recommendation:** 3
**Confidence:** 3

**Summary:**

This paper proposes SelfJudge, a self-supervised framework for judge verification in speculative decoding. The method trains verifiers by measuring semantic preservation through likelihood differences between original and token-substituted responses. Evaluations across benchmarks like GSM8K, MMLU, and CNN/DailyMail demonstrate that SelfJudge achieves superior accuracy-efficiency trade-offs compared to baselines. This idea is not novel, but the experiment and analysis is solid enough.

**Compliance With Llm Reviewing Policy:**

Affirmed.

**Final Justification:**

It has addressed part of my concerns. However, I still believe that the choice of hyperparameters is quite critical to the method, and this issue has not been fully resolved. I will revise my score slightly.

**Key Questions For Authors:**

1. Would the architecture of the verifier affect its effectiveness?
2. The threshold $\tau$ in expriments is currently set using AutoJudge labels on GSM8K. How should we determine this threshold in a purely self-supervised setting without relying on external baselines?
3. Can a verifier trained on one model family transfer to another?

**Limitations:**

please see Weaknesses.

**Strengths And Weaknesses:**

### Strengths
1. The proposed self-supervised training method effectively eliminates the dependency on human annotations or verifiable ground truth. This addresses the limitation of prior judge decoding approaches, enabling broader applicability across diverse NLP tasks.
2. The paper is well-written. It clearly identifies the datasets and model architectures used, facilitating replication of the experimental setup. Key hyperparameters, such as temperature, suffix length, and regularization ranges, are explicitly stated in the text and appendices. The authors detail the computational environment, including specific GPU types (V100, A100, H100) and parallelism strategies. The implementation of the proposed method and baselines within the vLLM framework is described with sufficient technical detail.

### Weaknesses
1. The method requires setting specific thresholds for data labeling and inference. This method may conflate stylistic, formatting, or model-specific preferences with semantic constraints.
2. The proposed verifier is distilling the target model’s local preferences over its own outputs, rather than learning an independent and generalizable criterion for semantic preservation.
3. The claim of applicability to “general NLP tasks” is relatively overstated.

---

> ### Author Rebuttal · Authors · 2026-03-30
>
> ### W1. Clarification on Thresholding with $\tau$ and $\theta$.
> ---
>
> We understand the reviewer’s concern regarding the reliance on specific thresholds for data labeling $\tau$ and inference $\theta$. However, note that rather than performing an exhaustive hyperparameter search, we focused on selecting these hyperparameters in a highly conservative approach to ensure practicality of SelfJudge.
>
> **1. Selecting $\tau$**. For practicality, $\tau$ is set once in a principled manner, not arbitrarily tuned. In **W2&Q2** of [Reviewer eQ6q](https://openreview.net/forum?id=9DAX4oLVhy&noteId=O5anWjgJXq), we investigate the sensitivity of SelfJudge to $\tau$, providing that **1) setting $\tau$ is not difficult thanks to its robustness** and **2) SelfJudge can achieve much better performance than current AutoJudge-based selection** if we slightly tune $\tau$.
>
> **2. Selecting $\theta$.** Our primary design principle was to minimize the acceptance of semantically incorrect tokens. Hence, we set $\theta$ based on the best recall and F1 scores. As shown in our main results, operating at the best-recall threshold accelerates inference with near-zero degradation in task performance. Furthermore, the sensitivity analysis in Figure 3 demonstrates that SelfJudge consistently outperforms baselines across a wide range of $\theta$ values. This robustness indicates that performance is not contingent on fine-tuned threshold selection
>
> **Regarding the conflation of style and semantics.** our method intentionally permits minor stylistic, and formatting because they do not alter the core semantic meaning. Allowing these benign variations is exactly what enables us to improve inference speedup without compromising output quality.
>
>
> ---
> ### W2: Target model distillation as the principled criterion for semantic preservation
> ---
>
> We argue that distilling the target model's preferences is not a limitation but the correct design choice for speculative decoding. The fundamental objective of SD is to preserve the target model's generation quality. Therefore, the relevant notion of "semantic preservation" is inherently defined with respect to the target model: a substitution is acceptable if and only if it does not degrade the target model's intended output.
>
> Our results confirm that this target-grounded criterion is in fact more generalizable than external alternatives. As shown in Table 1, a single verifier trained on GSM8K, LiveCodeBench, and Dolly15K generalizes effectively to unseen tasks including MMLU and CNN/DailyMail. In contrast, AutoJudge, which relies on the external criterion of answer correctness, shows poor generalization to these out-of-domain tasks. This directly demonstrates that the target model's own likelihood-based assessment captures a transferable notion of semantic equivalence, whereas domain-specific ground truths do not.
>
> ---
> ### W3: Generalizability of SelfJudge
> ---
>
> We acknowledge that "general NLP tasks" could be interpreted more broadly than our current evaluation covers. That said, we believe our experimental scope represents **a meaningful advance compared to prior judge verification methods.**
>
> **Broader coverage than existing methods.**  AutoJudge can only generate training labels for tasks with verifiable final answers, and accordingly shows poor generalization to unseen domains. As shown in Table 1 and Table 4, it achieves minimal speedup on MMLU and significant degradation on LiveCodeBench. In contrast, SelfJudge achieves consistent speedups across all five evaluation tasks from a single verifier trained on GSM8K, LiveCodeBench, and Dolly15K, including held-out tasks (MMLU, CNN/DailyMail) without any task-specific tuning.
>
> **Additional experiments on unseen domains.** To further validate generalizability of SelfJudge, we conduct experiments on LegalBench [1] and MedQA [2] — domains entirely absent from verifier training. Please kindly refer to W3&Q3 of Reviewer eQ6q.
>
> This result indicates that our approach **generalizes well even if the verifier has never seen the domain** like Legal and Medical questions.
>
> ---
> ### Q1:  Investigate Verifier with Complex Architecture & Q2: Regarding $\tau$ Selection
> ---
> Please refer to W1&Q1 and W2&Q2 of Reviewer eQ6q, respectively.
>
>
> ---
> ### Q3: Transferability of Verifier
> ---
>
> In our current experiments, we train separate verifiers for each model family (Llama-3 and Qwen-2.5), as the hidden representation spaces differ across architectures in both dimensionality and learned feature structure. Hence, direct transfer of verifiers between different model families is not feasible.
>
> However, training a per-model verifier on top of this existing pair introduces small additional cost. SelfJudge generates 53K labels in only 8.5 hours even for a 70B model (vs. 120 hours for AutoJudge), and the logistic regression training itself completes in seconds.
>
> [1] https://huggingface.co/datasets/nguha/legalbench
> [2] https://huggingface.co/datasets/bigbio/med_qa

---

> > ### Author Rebuttal · Reviewer_7AMQ · 2026-04-04
> >
> > Thank you for the response. It has addressed part of my concerns. However, I still believe that the choice of hyperparameters is quite critical to the method, and this issue has not been fully resolved. I will revise my score slightly.

---

### Official Review · Reviewer_ZFDW · 2026-03-13

**Soundness:** 2
**Presentation:** 2
**Significance:** 2
**Originality:** 2
**Overall Recommendation:** 4
**Confidence:** 3

**Summary:**

SelfJudge proposes a self-supervised judge-verification framework for speculative decoding, where the target model itself generates training labels by measuring whether token substitutions preserve the semantics of the original response, rather than relying on human annotation or task-specific verifiable answers.
Its main contributions are a semantic-preservation score for automatic verifier training and a lightweight context-aware verifier that improves the accuracy-efficiency trade-off across diverse NLP tasks, outperforming prior judge-decoding baselines in the reported experiments.

**Compliance With Llm Reviewing Policy:**

Affirmed.

**Final Justification:**

I thank the authors for their rebuttal. They have provided extensive clarifications addressing the concerns I raised about the paper, and my questions have now been fully resolved.

**Key Questions For Authors:**

When the judge verifier accepts tokens that the alignment-based verifier would reject, the resulting generation distribution is no longer identical to the standard target sampling distribution. Could the authors precisely define what is meant by “maintaining the distributional guarantees of alignment-based verification”? Does this refer to a strict distributional guarantee, or simply to the fact that the alignment verifier still acts as a fallback mechanism?
The calibration of τ\tauτ relies on unacceptable tokens identified by AutoJudge on GSM8K. Is there a pure SelfJudge calibration procedure that does not depend on AutoJudge? Additionally, how sensitive is τ\tauτ to the task domain, model family, and draft/target model pair?

**Limitations:**

yes

**Strengths And Weaknesses:**

Strengths
1. The proposed SelfJudge used during inference is lightweight and practical, and appears to exhibit good generalization properties.
2. As shown in Table 1, SelfJudge achieves a better overall speed–quality trade-off compared to standard SD and the AutoJudge family, rather than only showing occasional gains on a specific task.

Weaknesses
1. There is a noticeable mismatch between the training objective and the information available at inference time, but the paper does not analyze this sufficiently. The training labels are generated using bidirectional context (prefix + suffix), whereas at inference time the verifier cannot observe future tokens and must make online predictions based only on the current hidden state. While the paper demonstrates that labeling with future context leads to better supervision, it does not establish to what extent a forward-only verifier can realistically recover this information.
2. A key property of standard speculative decoding (SD) is that rejection sampling preserves the target distribution. Earlier in the paper, this property is also emphasized when discussing alignment-based verification. However, in the two-stage setup of SelfJudge, a token is accepted as long as it passes either the judge or the alignment verifier. Consequently, the judge may accept tokens that rejection sampling would otherwise reject. Strictly speaking, this no longer preserves the original target distribution. Therefore, the claim that the method operates “while maintaining the distributional guarantees of alignment-based verification” appears questionable from a theoretical standpoint, or at least requires a more precise definition of what “maintain” means in this context.
3. Although the paper emphasizes that the method does not rely on human supervision or verifiable ground truth, the threshold τ\tauτ is not fully endogenous. The authors set τ\tauτ using 100 GSM8K samples and the score quantiles of tokens labeled as unacceptable by AutoJudge, ensuring high recall on answer-critical tokens. This weakens the claim of being purely self-supervised, and raises concerns that the threshold may be implicitly anchored to a particular task domain.

---

> ### Author Rebuttal · Authors · 2026-03-30
>
> ### W1: Clarification on Training Objective and Inference Mismatch
> ---
>
> We respectfully note that this is not a mismatch but an intentional design: we leverage bidirectional context solely to produce higher-quality training labels, while the verifier learns to predict these labels from forward-only hidden states $h_t$​. The key question is whether $h_t$​ carries sufficient information to predict these labels and our ablation directly answers it.
>
> To directly measure this, we train verifiers with varying suffix lengths N ∈ {0, 5, 10, 20, 50, 100} used during label construction, where the verifier's input is hidden state $h_t$.
>
> Dataset| N | 0 | 5 | 10 | 20 | 50 | 100 |
> |---|:---:|:---:|:---:|:---:|:---:|:---:|:---:|
> |GSM8k| m |10.2|10.5| 10.6 | 11.3 | 10.8 |10.6|
> |GSM8k| Acc. | 80.2 | 80.4 | 80.4 | 80.5 | 80.4| 80.2|
> |MMLU| m | 4.7 | 4.8 | 4.9 | 5.4 | 5.2 | 4.8 |
> |MMLU| Acc. | 64.6 | 64.4 | 64.5 | 64.3 | 64.3 | 64.2|
>
> - At N=0 (prefix-only labels), the verifier shows limited improvement over standard SD, as its acceptance criterion reduces to a simple confidence threshold similar to rejection sampling.
> - Compared to N=0 (prefix-only), training with N=20 improves accepted length on GSM8K and MMLU while maintaining accuracy, demonstrating that forward-only hidden states can substantially recover the bidirectional signal.
> - Beyond N=20, performance slightly degrades, suggesting that very long suffixes introduce noise rather than useful signal. This directly establishes that forward-only hidden states can effectively recover the bidirectional signal when trained with appropriate supervision.
> - We additionally measured the AUC score of the verifiers N=0 (AUC: 66.3) and N=20 (68.3), implying that hidden states can recover the forward-only labels, and bi-directional labels in a similar-level.
>
> **Why this works.** Recent work has shown that autoregressive hidden states encode predictive information about future token trajectories, not just prefix summaries [1, 2]. Moreover, the hidden state includes a self-correction signal if a generated token is incorrect [3]. Our bidirectional labels provide the supervision needed to extract this latent information, effectively distilling sequence-level semantic judgments into a simple forward-only classifier.
>
>
>
> ---
> ### W2: Clarification on distributional guarantees in two-stage verification
> ---
>
> We agree that the phrase "maintaining the distributional guarantees" is imprecise, as the two-stage verification allows tokens that rejection sampling would reject. We will revise this to clarify that our method trades exact distributional equivalence for efficiency while strictly preserving semantic fidelity.
>
> However, to clarify, the two-stage paradigm (alignment + judge), introduced by [3], intentionally relaxes this constraint by accepting tokens that rejection sampling would reject if they are semantically compatible. This is not a side effect but **the core design goal of JudgeDecoding and its literature** — replacing strict distributional equivalence with semantic equivalence to unlock greater speedup.
>
> Hence, we rigorously report task performance alongside efficiency, confirming that SelfJudge achieves the smallest performance degradation among all judge methods, proving our semantic preservation approach is effective.
>
>
> ---
> ### W3: Clarification on the calibration of $\tau$ and its domain dependency
> ---
>
> We address two points: (1) whether $\tau$ can be calibrated without AutoJudge, and (2) how sensitive $\tau$ is to the calibration source.
>
> **$\tau$ calibration without AutoJudge.**. Note that our $\tau$ setting based on AutoJudge is **not chosen for optimal performance, but for practicality**—it reduces the cost of parameter search. To set $\tau$ in a fully self-contained manner, we propose and validate a simple quantile-based calibration that requires no external method or ground truth: given the distribution of semantic preservation scores computed by SelfJudge itself across training data (GSM8K train + LiveCodeBench train + Dolly15k), **we can select $\tau$ as a quantile of this distribution.**
>
> To confirm that selecting $\tau$ is reliable, we provide a sensitivity analysis on $\tau$ by varying it according to quantiles $\{10, 30, 50, 70, 80, 90\}$. Due to the characters limit of ICML rebuttal, please kindly refer to **W2&Q2** of [Reviewer eQ6q](https://openreview.net/forum?id=9DAX4oLVhy&noteId=bmRonTiuKy).
>
> Given these results, we propose the endogeneous calibration method at a moderate quantile (e.g., Q30–Q90) of the score distribution. This removes all dependency on AutoJudge or any task-specific ground truth while preserving stable performance.
>
>
> [1] Pal et al. (2023). Future Lens: Anticipating Subsequent Tokens from a Single Hidden State.
>
> [2] Wu et al. (2024). Do Language Models Plan Ahead for Future Tokens?
>
> [3] Bachmann, G., et al. (2025). Judge Decoding: Faster Speculative Sampling Requires Going Beyond Model Alignment. ICLR 2025.

---

> > ### Author Rebuttal · Reviewer_ZFDW · 2026-04-03
> >
> > I thank the authors for their rebuttal. They have provided extensive clarifications addressing the concerns I raised about the paper, and my questions have now been fully resolved. And I will raise my score.

---

### Decision · Program_Chairs · 2026-04-30

**Decision:**

Accept (regular)

**Comment:**

The paper proposes SelfJudge, an extension of judge-based speculative decoding. It trains the judge in a self-supervised way using a score derived from the target model. Reviewers generally agreed that the idea is well motivated and that the empirical results are promising. The main concerns are on the reliance on threshold calibration, the gap between bidirectional labeling and forward-only inference, and the lack of distributional guarantees once the judge relaxes rejection sampling.